# First Report on Cyanotoxin (MC-LR) Removal from Surface Water by Multi-Soil-Layering (MSL) Eco-Technology: Preliminary Results

Roseline Prisca Aba [1,2], Richard Mugani [1,2], Abdessamad Hejjaj [2], Nelly Brugerolle de Fraissinette [3], Brahim Oudra [1,2], Naaila Ouazzani [1,2], Alexandre Campos [3], Vitor Vasconcelos [3,4], Pedro N. Carvalho [5] and Laila Mandi [1,2,*]

1   National Center for Studies and Research on Water and Energy, Cadi Ayyad University, P.O. Box: 511, 40000 Marrakech, Morocco; abaroselineprisca@gmail.com (R.P.A.); richardmugani@gmail.com (R.M.); oudra@uca.ma (B.O.); ouazzani@uca.ac.ma (N.O.)
2   Laboratory of Water, Biodiversity and Climate Change, Faculty of Sciences Semlalia, Cadi Ayyad University, P.O. Box: 2390, 40000 Marrakech, Morocco; a.hejjaj@uca.ma
3   CIIMAR, Interdisciplinary Center of Marine and Environmental Research, University Porto, 4099-002 Porto, Portugal; nelly.brugerolle@ciimar.up.pt (N.B.d.F.); acampos@ciimar.up.pt (A.C.); vmvascon@fc.up.pt (V.V.)
4   Department of Biology, Faculty of Sciences, University of Porto, 4099-002 Porto, Portugal
5   Department of Environmental Sciences, Aarhus University, 8000 Aarhus, Denmark; pedro.carvalho@envs.au.dk
*   Correspondence: mandi@uca.ac.ma

**Abstract:** Cyanobacteria blooms occur frequently in freshwaters around the world. Some can produce and release toxic compounds called cyanotoxins, which represent a danger to both the environment and human health. Microcystin-LR (MC-LR) is the most toxic variant reported all over the world. Conventional water treatment methods are expensive and require specialized personnel and equipment. Recently, a multi-soil-layering (MSL) system, a natural and low-cost technology, has been introduced as an attractive cost-effective, and environmentally friendly technology that is likely to be an alternative to conventional wastewater treatment methods. This study aims to evaluate, for the first time, the efficiency of MSL eco-technology to remove MC-LR on a laboratory scale using local materials. To this end, an MSL pilot plant was designed to treat distilled water contaminated with MC-LR. The pilot was composed of an alternation of permeable layers (pozzolan) and soil mixture layers (local sandy soil, sawdust, charcoal, and metallic iron on a dry weight ratio of 70, 10, 10, and 10%, respectively) arranged in a brick-layer-like pattern. MSL pilot was continuously fed with synthetic water containing distilled water contaminated with increasing concentrations of MC-LR (0.18–10 μg/L) at a hydraulic loading rate (HLR) of 200 L m$^{-2}$ day$^{-1}$. The early results showed MC-LR removal of above 99%. Based on these preliminary results, the multi-soil-layering eco-technology could be considered as a promising solution to treat water contaminated by MC-LR in order to produce quality water for irrigation or recreational activities.

**Keywords:** multi-soil-layering eco-technology; cyanotoxins; microcystin-LR; water treatment

## 1. Introduction

Cyanobacteria, also known as blue-green algae, have existed since about 3500 Ma. They are Gram-negative bacteria and are the source of life on earth through their production of oxygen [1,2]. Cyanobacteria occur naturally in freshwater [3,4], brackish, and marine water [5,6]. Due to anthropogenic activities and global warming, cyanobacteria can rapidly increase and generate bloom. Olokotum et al. [7] in their review described the multiple ways in which human growth, as well as human activities, is connected to the increasing occurrence of cyanobacterial blooms in Lake Victoria. There have been frequent reports on toxic cyanobacterial blooms in surface waters around the world [8–12].

Cyanobacterial blooms can be harmful to the environment, animals, and human health. Decrease in oxygen concentration, nauseating odors, water coloring, and toxicity are some of the ecological disturbances caused by cyanobacterial blooms [13]. The environmental problem of toxic cyanobacteria blooms lies in the fact that the resulting toxins, dissolved in water, can accumulate in the tissues of fish and other aquatic biotas. These accumulated toxins can be transferred to humans via the food chain [14]. Similarly, studies have shown that these toxins can also accumulate in the edible parts of plants [15,16]. In doing so, they become a health hazard for consumers of these unsafe products. Many genera of cyanobacteria can produce toxic secondary metabolites called cyanotoxins [4,17,18]. Cyanotoxins are essentially endotoxins that can be released into the environment after cell lysis. These toxins constitute a huge group of chemical compounds that differ in their molecular structure and toxicological properties [19,20]. They can be classified according to the lesions they cause in different organs of animals. This includes hepatotoxins or liver toxins (microcystins, cylindrospermopsins, and nodularins), neurotoxins, or nervous system toxins that target the neuromuscular junction (anatoxins and saxitoxins), dermatotoxins or skin toxins (lyngbyatoxins), irritant toxins (lipopolysaccharides), and cytotoxins (cell toxins). The presence of these toxins has been reported in waters around the world, including Europe [21], England [22], Turkey [23], Canada [24], China [25], Tanzania [26], Ethiopia [27], Tunisia [28], and Morocco [29].

The contamination of surface waters by cyanotoxins can lead to water quality problems for fisheries, aquaculture, and livestock farming as well as health risks for humans and animals [30]. Irrigation with water containing cyanobacterial toxins can inhibit plant growth [13]. The number of publications concerning phytotoxic effects of cyanotoxins on agricultural plants has increased [13,16,31,32]. Drinking water and recreational activities are also other ways by which humans may be exposed to cyanotoxins [14,33].

In terms of global impact on health and water quality, microcystins are the most indexed. Microcystins are cyclic heptapeptides. Around 250 variants have been identified in waters around the world [34]. MC-LR is the most toxic variant reported worldwide. MC-LR is the most abundant, exhibiting 46.0–99.8% of the total concentration of MCs in natural blooms [35]. Because of its acute and chronic toxicity, the WHO has set a tolerance margin for MC-LR of 1 µg/L in drinking water [36].

The occurrence of toxic cyanobacteria blooms has been confirmed in many water bodies in Morocco. More than 18 out of 26 lake reservoirs used for recreational and drinking water reservoirs in Morocco contained toxic cyanobacteria [37]. Toxicological studies of 19 toxic cyanobacteria strains isolated from reservoirs and ponds in Morocco showed concentration of MCs between 26.8 and 1884 µg/g dry weight [38]. Douma et al. [19] carried out a toxicological assessment in Moroccan inland waters that confirmed the presence of toxic strains and five MC variants (MC-RR, MC-LR, MC-YR, MC-WR, and MC-FR). This contamination constitutes a real threat to human and environmental health. In Dayet-Aoua lake, Morocco, the presence of various toxic MC congeners was confirmed with high toxin concentration (185.56 $\mu g \, g^{-1}$ dry weight) [17]. Molecular analysis showed *Microcystis aeruginosa* as the species responsible for most of the microcystins (3240 µg $g^{-1}$ dry weight cyanobacterial biomass) that occur in the Moroccan Lalla Takerkoust reservoir. In this reservoir, microcystins can persist throughout the year [39]. Beyond planktonic cyanobacterial toxic blooms such as those caused by *Microcystis aeruginosa*, the occurrence of toxic benthic *Nostoc* producing MC-LR and equivalent with an estimated concentration of 139 $\mu g.g^{-1}$ dry weight has been reported in Moroccan freshwater [40]. Although both the *Nostoc* and *Microcystis* genera contain toxic species, the secreted toxins are not always the same. Toxic species in the genus *Microcystis* secrete mainly MCs, while toxic species in Nostocaceae, such as *Nostoc punctiforme* and *Nostoc muscorum*, can produce both nodularins [41] and MCs [19]. In addition, Wannicke et al. [41] showed that Nostocaceae, which are heterocystous filamentous cyanobacteria in contrast to *Microcystis* genus, which are colonial species, produce more nodularins under nitrogen replete and diazotrophic conditions.

Cyanobacterial bloom is a major problem, especially in arid and semiarid countries that have no alternative but to use surface water. Given the impact of this phenomenon on a global scale, the need to treat water contaminated by cyanotoxins is obvious. A variety of methods, such as adsorption [42,43], ultrasonic technology [44], electrocoagulation–Fenton [45], osmose reversible and photocatalysis [46,47], ultraviolet radiation and oxidation [48–52], and nanotechnology [53], are used to eliminate cyanobacteria and their toxins in water. However, the conventional water treatment methods are expensive, consume a lot of energy, and require specialized personnel and equipment. On the other hand, environmental factors are critical in the choice of technology for the treatment of cyanobacterial bloom. Naturally, bacteria such as *Sphingomonas sp*, *Bacillus sp*, *Paucibacter toxinivorans*, *Sphingosinicella microcystinivorans*, and *Pseudomonas aeruginosa* can degrade cyanobacteria and cyanotoxins by using them as a source of nutrients [54–59]. Regarding cyanobacterial degradation, Ndlela et al. [60] established a chronology of reports dealing with this topic from 2000 until 2017. The authors identified the main mechanisms used by bacteria to degrade cyanobacteria through algicidal, lytic, or growth inhibitory activities. In another study, Ndlela et al. [61] showed that the addition of isolate 3y to a *Microcystis aeruginosa* culture caused the cells to be deflated compared to the uninoculated control. The authors hypothesized that the decrease in cell number as well as the deflated form of *Microcystis aeruginosa* cells was because the cells were stressed and dying due to the addition of the bacterial isolate. In addition to bacteria, other biological agents attack cyanobacteria. Fungi present themselves as good candidates. *Phanerochaete chrysosporium*, for example, can destroy algal cells as well as reduce the expression of the *Microcystis aeruginosa* toxin gene [62]. Likewise, Han et al. [63] documented *Microcystis aeruginosa* membrane decomposition abilities by endopeptidase enzymes and polysaccharide lyases from *Bjerkandera adusta* and *Trametes versicolor*. Furthermore, the lytic role of cyanophages directed against *Microcystis aeruginosa* has been demonstrated [64]. Biodegradation of MC-LR using biologically active slow sand filter as a low-cost water treatment technology was experimented by Bourne et al. [58]. Their result showed its effectiveness for the removal of MCs in water. However, the large-scale application of this system needs more investigation. Moreover, coagulation-flocculation technology using natural coagulants (*Vicia faba* seeds and *Opuntia ficus indica* cladodes) was demonstrated to be an eco-friendly and low-cost technology to reduce cyanobacterial toxic blooms. These two natural coagulants were able to reduce the turbidity of water, chlorophyll a, and carotenoids by up to 85% [65]. In another study, Álvarez et al. [66] performed coagulation-flocculation to freshwater algae using 10 mg/L of *Pinus pinaster* bark, which resulted in removal efficiency of 68.10%. However, this technology has the disadvantage of releasing intratoxins that can pose problems for water use. Bavithra et al. [67] evaluated the potential ability of constructed wetland (CWs) to remove cyanobacteria and MC-LR from freshwater. The authors found that constructed wetland can remove 94% of *M. aeruginosa* and around 99% of MC-LR in just a one-week treatment cycle. Despite CWs having the capability to treat eutrophic waters, this system is known to occupy considerable land space.

Recently, multi-soil-layering (MSL) system, a natural and low-cost technology, has been introduced as an attractive cost-effective, and environmentally friendly technology that can be an alternative to conventional water treatment methods [68,69]. The MSL system has high hydraulic loading rates and small land space requirements, which are ideal for application. The MSL system has shown good performance in the reduction of organic matter, nutrients, and pathogens from wastewater [68,70]. In previous studies, the MSL system has been successfully used for the treatment of domestic wastewater [71], polluted river water [72], livestock wastewater [73], dairy effluent [74], leachate [75,76], and olive mill wastewater [77].

However, to our knowledge, the MSL system has never been used to treat cyanobacterial harmful algae bloom and microcystin-LR. Therefore, this study aimed to investigate, for the first time, the removal of microcystin-LR by low-cost MSL eco-technology.

## 2. Materials and Methods

### 2.1. MSL Pilot Description

The lab-scale MSL system made of a rectangular glass tank was designed with the dimension of 60 × 10 × 70 cm (L × W × H). The MSL system pilot was composed of soil mixture layers (SML) and permeable layers (PLs) that were arranged in a brick-layer-like pattern. A pozzolan with a size of 3.5–5 mm was filled as PLs. The SML consisted of a mixture of local sandy soil, iron metal, charcoal, and sawdust on a dry weight ratio of 70, 10, 10, and 10%, respectively, and was filled as anaerobic layers. A perforated aeration pipe was installed in between the third and the fourth soil mixture layers, which was about the middle of the MSL system, for uniform diffusion of air. An aeration pipe can control the aerobic and anaerobic profile of the MSL by sending air from outside the system. A full description of the MSL pilot is presented in Figure 1.

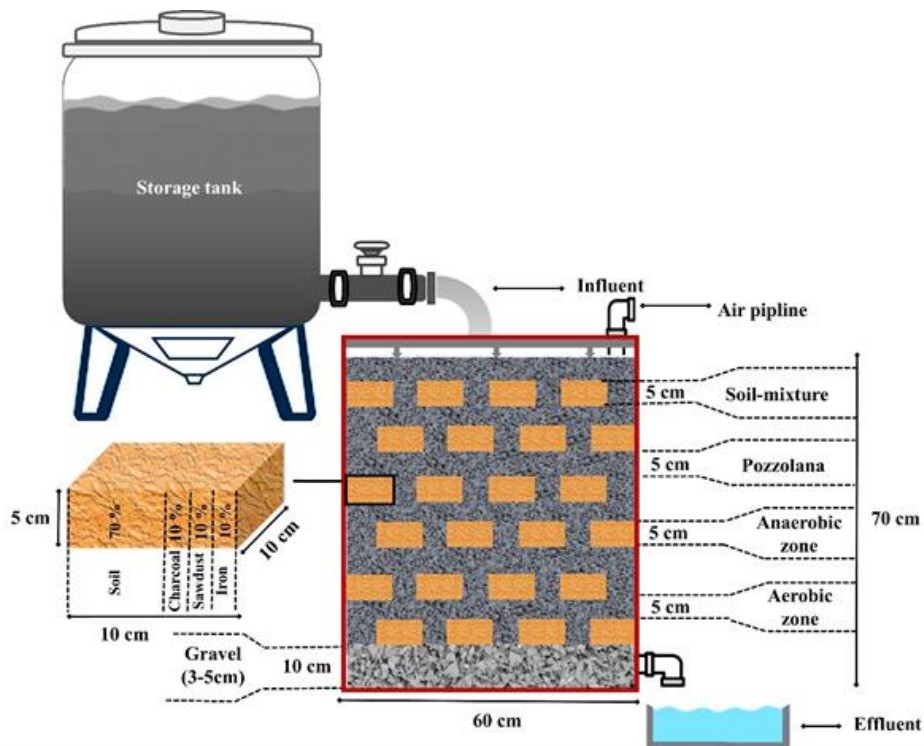

**Figure 1.** Scheme of Multi-Soil-Layering (MSL) pilot plant.

### 2.2. MSL Experimental Conditions and Influent Water Quality

The MSL system was initiated in September 2019 and continuously operated for six weeks. Acclimation of the MSL pilot system was done with pH 7 neutralized distilled water for two weeks. The influent was synthetic water created by contaminating distilled water with increasing concentrations of microcystin (MC-LR) over the weeks. The distilled water was neutralized to pH 7. The extraction and purification of microcystin-LR from the 2018 lyophilized cyanobacterial bloom was performed according to the method of Fastner et al. [78]. The biological material used to prepare the microcystin concentrations was a bloom collected at the Lalla Takerkoust dam, Marrakech (Morocco), in 2018 and freeze-dried to powder form. This bloom, whose enzyme-linked immunosorbent assay (ELISA) characterization shows a total microcystin concentration of 4147.8 µg/g of dry matter, was supplied to us already characterized and ready for use by the Laboratory of Water, Biodiversity, and Climate Change at the Faculty of Sciences Semlalia, Cadi Ayyad University (Marrakech, Morocco).

Therefore, five different microcystin concentrations were prepared from the freeze-dried 2018 bloom deriving from Lake Takerkoust. The various MC-LR equivalent con-

centrations used were 0.18, 0.91, 2.5, and 5 µg/L. Feeding duration was one week per concentration. Based on the hydraulic flow rate of 200 L m$^{-2}$ day$^{-1}$, daily intake was estimated to be approximately 12 L/day, equating to 84 L/week. The hydraulic flow rate was adjusted by a Masterflex® L/S® series peristaltic pump. For the first three concentrations (0.18, 0.91, and 2.5 µg/L), the influent was prepared daily, while for the last concentration (5 µg/L), the entire weekly effluent was prepared on the same day.

### 2.3. Samples Collection and Processing

Sampling of influent and effluent water of the MSL system was done three times a day (9 a.m., 1 p.m., and 5 p.m.) for the first three concentrations of MC-LR (0.18, 0.91, and 2.5 µg/L). Only an average sample of each day was submitted for quantification of MC-LR in triplicate. For the last concentration (5 µg/L), samples were collected three times a day (9 a.m., 1 p.m., and 5 p.m.) on the first and last days of feeding. Physicochemical parameters such as pH, dissolved oxygen (DO), electrical conductivity (EC), and total dissolved solids (TDS) were measured immediately on the samples using a Hanna HI 9829 multiparameter probe. Samples for the determination of microcystin were collected with glass bottles, which were wrapped in aluminum foil and stored at −20 °C. These samples were then filtered through Whatman microfiber glass filter paper GF/C, D = 47 mm, and prepurified. Prepurification was performed according to Triantis et al. [79]. Briefly, 500 mL of water filtered through the Whatman GF/C filter was passed through a C18 MERCK LiChrolut® RP-18 column after being conditioned with 5 mL of 100% methanol and 5 mL of ultrapure water. After passing the sample, the column was rinsed with 20% methanol, and then elution was carried out with 5 mL of 100% methanol. Quantifications of MC-LR in all samples were performed by high-performance liquid chromatography–mass spectrometer (HPLC–MS).

### 2.4. Detection and Quantification of MC-LR by LC–ESI-MS

The LC-MS system used to quantify MC-LR was a liquid-phase chromatograph Alliance e2695 HPLC system coupled with a triple quadrupole spectrometry detector (Micromass® Quattro micro TM API) with electrospray (ESI) interface. Chromatographic separation was achieved on C18 Hypersil Gold column (100 × 4.6 mm I.D., 5 µm, Thermo-Scientific, Waltham, MA, USA). The columns were kept at 35 °C during analysis. The injected volume was 10 µL in loop partial mode. Samples were injected in positive polarity mode in full scan (30–2000 *m/z*) and SIR of 2 channels (135 and 995.5 *m/z*). The standards and samples were injected in duplicate, and for each set of 10 samples, a blank and two standards of different concentrations were introduced. The standard solution of MC-LR was purchased from CIFGA S.A. (Spain, Batch n° 15-001) with a concentration of 10 µg/mL. The system was calibrated using 8 dilutions of the standard solution of MC-LR (between 0.01 and 2 µg/mL) diluted in 50% methanol (MeOH).

A gradient elution was used with ultrapure water (mobile phase A) and MeOH (mobile phase B), both of which were acidified with 0.1% formic acid (30% A and 70% B at 0 min, 50% A and 50% B at 10 min, returning to initial conditions at 15 min and equilibrating for 5 min). Under these conditions, the MC-LR retention time was 5.71 min. After the toxin elution step on the SPE column, the limits of detection (LOD; S/N = 3) and quantification (LOQ; S/N = 10) of MC-LR were 0.008 and 0.01 µg/mL, respectively.

Samples were quantified using SIR with 2 channels (135 and 995.5 *m/z*) and precursor ion (*m/z* 995), and MC-LR reference fragment ions with *m/z* values of 375, 553, 599, 866, and 977 were monitored in the full scan mode in order to validate the presence of the toxin.

### 2.5. Statistical Analysis

All measurements/analyses were conducted in triplicate. Means and standard deviations were calculated. Similarly, the MSL removal rates of microcystin-LR were computed. Finally, the significant differences ($p < 0.05$) of the parameters at the input and output of the system were evaluated through a nonparametric Mann–Whitney test.

## 3. Results

### 3.1. Physical and Chemical Characteristics of Influent and Effluent Waters

The average temperature during the whole study was 2 °C. The pH of distilled water used as an influent in this experiment was adjusted to 7.35. The Mann–Whitney comparison between the pH at the inlet and outlet of the MSL pilot plant showed a significant difference ($p < 0.00001$) at the 95% confidence interval. In fact, as shown in Figure 2, there was an increase in pH from the inlet to the outlet (7.46 ± 0.13 to 8.31 ± 0.07).

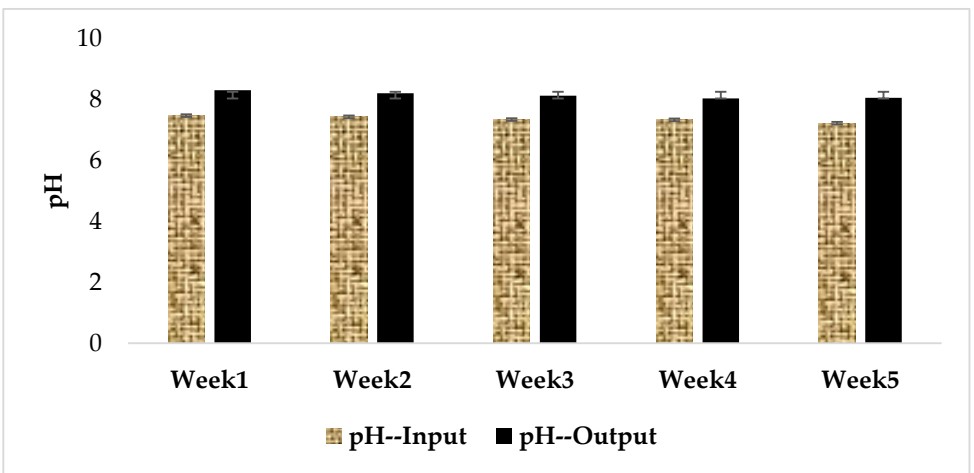

**Figure 2.** pH at the influent and effluent of the MSL pilot plant during the experiment.

The highest values of dissolved oxygen were 4.31 ± 0.20 mg/L and 4.81 ± 1.72 mg/L at the inlet and outlet, respectively, and the lowest values were 2.11 ± 0.12 and 2.13 ± 1.37 mg/L at the inlet and outlet, respectively. As shown in Figure 3, the Mann–Whitney test showed no significant difference between the input and output of the MSL system.

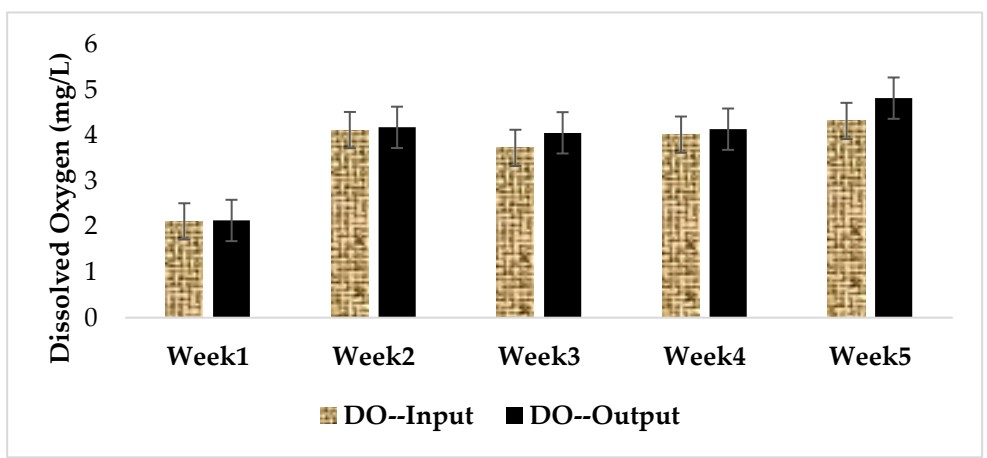

**Figure 3.** Dissolved oxygen (DO) of the influent and effluent during the experiment.

The highest values of electrical conductivity were 135.67 ± 129.65 μs/Cm and 315.17 ± 87.49 μs/Cm at the inlet and outlet of the MSL pilot, respectively, and the lowest values were 17.69 ± 7.47 μs/Cm and 215.17 ± 16.24 μs/Cm at the inlet and outlet, respectively, of the MSL (Figure 4).

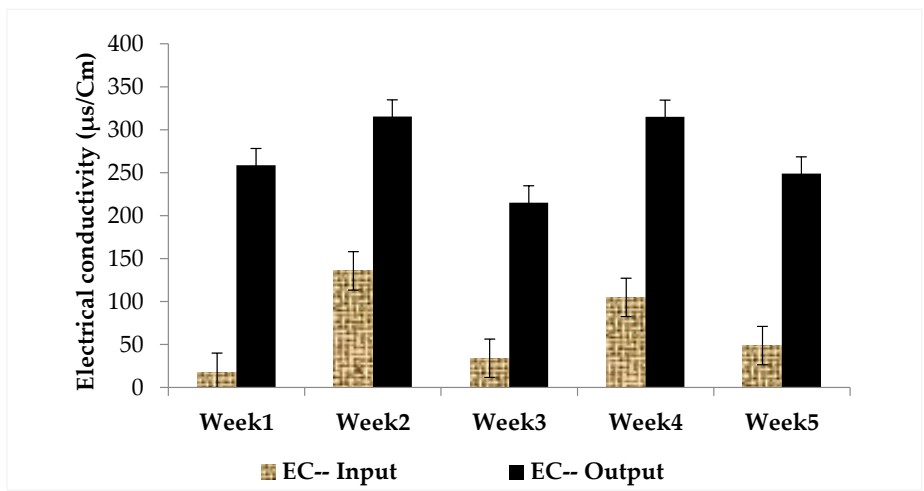

**Figure 4.** Electrical conductivity of the influent and effluent during the experiment.

The highest values of total dissolved solids (TDS) were 65.81 ± 61.77 mg/L and 158.18 ± 43.70 mg/L at the inlet and outlet of the MSL pilot, respectively, and the lowest values were 9.38 ± 2.97 mg/L and 108.33 ± 11.17 mg/L at the inlet and outlet of the MSL pilot, respectively. Furthermore, the Mann–Whitney test found significant differences (Figure 5) between the inlet and outlet of this parameter ($p < 0.0001$) at the 95% confidence interval.

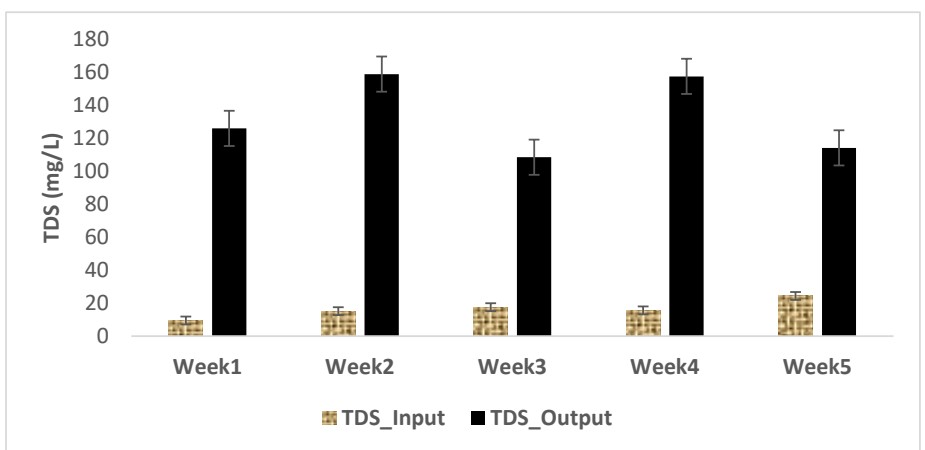

**Figure 5.** Total dissolved solids of the influent and effluent of the MSL pilot plant during the experiment.

### 3.2. Quantification MC-LR

The results of the MC-LR quantification are shown in Figure 6.

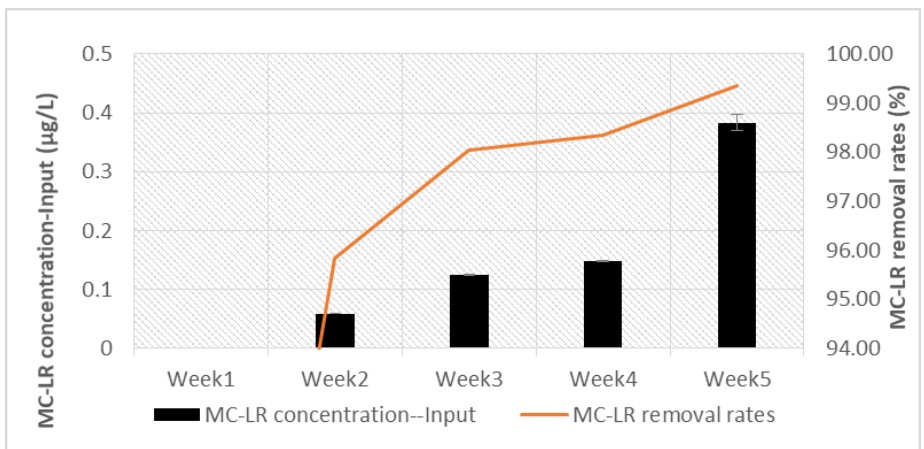

**Figure 6.** MC-LR removal rates by multi-soil-layering eco-technology during the experiment.

Because the bloom used was characterized by the ELISA method, we worked with total microcystin concentrations. The analysis of water toxins was carried out at the Interdisciplinary Center of Marine and Environmental Research (CIIMAR), Portugal, by LC–ESI-MS using a single standard MC-LR. Thus, the concentrations of MC-LR at the inlet and outlet were much lower than those of the microcystin equivalents used to prepare the distilled water contamination solutions.

Across the experiment, regardless of the dose used in different weeks, the Mann–Whitney input/output test showed a significant difference with $p < 0.00001$ at the 95% confidence interval. In short, in all cases, the toxin concentration at the MSL outlet was below detection (Table 1).

**Table 1.** MC-LR concentrations in the influent and effluent of the MSL system (mean and standard deviation, $n = 3$) and removal rates.

| Week | Concentration of the Extract in MC Equivalent (µg/L) at Input | Concentration of Extract in MC-LR (µg/L) at Input | MC-LR Concentration (µg/L) at Output | MC-LR Removal Rates (%) |
|---|---|---|---|---|
| 1 | 0.18 | nd * | nd * | - |
| 2 | 0.91 | 0.0585 (0.0001) | 0.0024 (0.0001) | 95.83% |
| 3 | 2.5 | 0.1248 (0.0001) | 0.0024 (0.0001) | 98.05% |
| 4 | 5 | 0.1479 (0.0014) | 0.0024 (0.0001) | 98.35% |
| 5, the first day of the week | 10 | 0.3832 (0.014) | 0.0024 (0.0001) | 99.36% |
| 5, the seventh day of the week | | 0.3765 (0.0012) | 0.0024 (0.0001) | 99.35% |

nd: not detectable, * below limit of detection.

Gradually, different concentration of microcystin up to 10 µg/L was injected to influent water for five weeks. The results of the microcystin quantification showed 95–99% removal rates of MC-LR from week one to week five. The microcystin removal rate was more than 95% from the first week.

## 4. Discussion

The present study investigated, for the first time, the potential of multi-soil-layering (MSL) eco-technology to remove MCs from surface water. Effluent water exhibits alkalinity. The alkalinity of treated water can be due to hydroxylation during the degradation of MC molecules. Indeed, the amino acids are detached from the adda part, making the medium alkaline. Kumar et al. [80] in their review reported that the degradation rate of microcystin increases in a medium of pH ≤8. In another study, Santos et al. [81] evaluated the influence

of acidity on the adsorption/desorption of MC-LR on sediments. In their study, the authors showed that at a temperature of 25 °C, pH 5 and 8 did not have a significant influence on the adsorption/desorption processes. Similarly, Terin et al. [82] in their study on a domestic slow sand filter obtained microcystin reduction to a concentration below 1.0 μg/L when the pH was 6.7. This alkalinity could also be due, in part, to the leaching of minerals (ions and solids) from the substrate of the MSL system. The increase in TDS and conductivity at the outlet of the MSL would be evidence of this leaching.

The increase in dissolved oxygen at the exit of the system is a sign of oxygenation in the MSL pilot system, the presence of microorganisms, biofilm formation, and therefore proper system operation. Indeed, this is a sign of good system performance as long as the system is ventilated. Thus, aerobic microorganisms can form a biofilm in all layers of the pilot and carry out the degradation of MCs without any constraint as to the presence of oxygen in the deep layers because there is always oxygenation.

The increase in electrical conductivity is contradictory to the results of other MSL systems treating domestic wastewater and polluted river water [72,83]. This could be explained by the fact that the MSL system only treated distilled water contaminated with MCs. There was no pollutant load in the feed water, hence the low electrical conductivity at the inlet of the MSL system. The leaching of soluble elements from the MSL substrate by distilled water and degradation of microcystin-LR would be the reason for the high electrical conductivity at the outlet of the pilot system.

On the other hand, the TDS values confirm electrical conductivity. Dissolution of inorganic salt compounds and organic matter in the MSL filter materials resulted in increased electrical conductivity and TDS. Nevertheless, these values still comply with the WHO recommended standards for drinking water.

During the last week (week 5), samples were collected on days 1 and 7. A slight decrease in the toxin concentration, from 0.3832 to 0.3765 μg/L, was noted. This decrease in the feed tank during the same week could be related to the adaptation and development during the five weeks of the bacteria capable of degrading MCs. If a bacterial community capable of managing MCs begins to appear in the feed tank during the fifth week, this could be an indication of a quasi-colonial presence of these microcystin-degrading bacteria in the layers of the MSL. This is because the system is designed to contain the nutrients and conditions favoring the formation of biofilm to accelerate treatment. This slight decrease in microcystin concentration from the inlet to the storage tank could also be due to the fact that PVC has the ability to adsorb MCs [84].

Adsorption, infiltration, and biodegradation are the major processes occurring in the MSL system for pollutant removal [85].

The removal of MCs is therefore enhanced by adsorption onto the porous pozzolan material. This is in agreement with several authors who have shown that filtration is one of the processes for removing microcystin [82,86,87]. However, biodegradation has been shown to be the major degradation process for MCs in slow sand filtration [56,82,88]. In the present study, the increase in the concentration of MCs and equivalent had no impact on the purification capacity of the MSL system. Several authors have shown that biofilm formation requires 4–7 months on sterile material [82,86,89]. This leads us to believe that this purification performance is also due to the formation and maturation of microorganisms capable of degrading the toxin. The dissolved oxygen at the MSL pilot outlet gradually increased, which is evidence of the aeration of the system and good microbial functioning, leading to the maturation of the biofilm inside the MSL substrate. Maintaining the removal rate above 99% throughout the experiment involves the combination of adsorption and biodegradation. Filtration and biodegradation would therefore be effective in the MSL system and are the major processes in the degradation of MCs.

## 5. Conclusions

The present work evaluated, for the first time, the efficiency of multi-soil-layering eco-technology to remove cyanotoxins from distilled water contaminated by MC-LR. Pre-

liminary results showed a very good capacity of the MSL system to eliminate MC-LR, with the removal rate reaching above 99%. The main mechanisms involved in this removal are probably infiltration, adsorption, and degradation. Therefore, the multi-soil-layering eco-technology could be considered an efficient and promising nature-based solution for the removal of cyanotoxins (MC-LR) from contaminated surface water. However, further research regarding the long-term cyanotoxin removal capacity of the MSL eco-technology and the mechanisms involved is needed before its potential application.

**Author Contributions:** Investigation, visualization, data analysis, writing—original draft, R.P.A., and R.M.; resources, methodology, investigation, A.H.; investigation, N.B.d.F.; methodology, investigation, supervision, resources, writing—review and editing, N.O. and B.O.; writing—review and editing, A.C. and P.N.C.; project administration, resources, A.C. and V.V.; funding acquisition, resources, supervision, investigation, writing—review and editing, L.M. All authors have read and agreed to the published version of the manuscript.

**Funding:** This research has received funding from the European Union's Horizon 2020 research and innovation program under the Marie Skłodowska-Curie grant agreement No 823860.

**Institutional Review Board Statement:** Not applicable.

**Informed Consent Statement:** Not applicable.

**Data Availability Statement:** The data presented in this study are available in this article.

**Acknowledgments:** The authors would like to thank the National Center for Studies and Research on Water and Energy (Cadi Ayyad University, Morocco) and the Interdisciplinary Centre of Marine and Environmental Research (CIIMAR) of the University of Porto (Portugal) for their scientific and technical support to this work.

**Conflicts of Interest:** The authors declare no conflict of interest.

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
