# Peer review of "First Report on Cyanotoxin (MC-LR) Removal from Surface Water by Multi-Soil-Layering (MSL) Eco-Technology: Preliminary Results"

_water, doi:10.3390/w13101403_

Round 1
Reviewer 1 Report
The aim of this study was to investigate the removal of microcystin-LR (their particularly high concentrations occurring with intense blooms of cyanobacteria) by low-cost eco-technology called Multi-Soil-Layering (MSL).
After extending the research on this topic (e.g. a case study for selected rivers, lakes and other water bodies with troublesome cyanobacterial blooms occurring in them) and confirming the conclusions, the presented method of removing cyanotoxins could be applied for the treatment of surface waters.
The subject of this project important and interesting but there are several problems in the manuscript need to be clarified and revised before the manuscript can be considered for publication.
Although the topic taken up (First report on cyanotoxins (MC-LR) removal from surface water by multi-soil-layering (MSL) eco-technology: preliminary results) is very interesting, presented in a clear and understandable way, there were still some mistakes in it and fragments that require supplementing and explanation.
Here some comments and suggestions:
Introduction:
Line 59: …China [22,23]; [22] OK but [23]?, Compare it with line 75.
Line 64: … has recently increased [29–32]. "Recently" means the years 2011-2016 according to the authors?
Line 94: …oxidation [48–53] and Nanotechnology [53] = …oxidation [48–52] and nanotechnology [53].
Materials and methods
Figure 1. - I have doubts whether the aerobic zone layers and the pozzolana were marked / described in the right place.
- 145: ELISA (enzyme-linked immunosorbent assay ?) - Please expand the abbreviation.
Results
In Figure 1-5- No 0Y axis is drawn.
Pay attention to the scale / enlargement of figures - it should be the same (especially in contrast to fig. 3).
Line 231 – Fig. 5. - The caption under the drawing should be corrected (TDS!)
Line 234 – Highlighted “Figure 6” (?)
Line 238 – CIIMAR - enter the full name of the institution.
Line 247 - Correct the caption under table 1, include "*"
Discussion
Line 260: pH=5 and pH=8
Line 262: pH=6.7
References
Please - number each item in your bibliography list.
Provide non-English items also translated into English (applies to items 11 and 34).
In [78] “e ffi cient” = “efficient”
Items [78] and [81] share the same title.
Line 522-524: Correct the title in position [81]
[81] Latrach, L.; Ouazzani, N.; Masunaga, T; Hejjaj,A.; Bouhoum, K.; Mahi, M.; Mandi, L. Domestic wastewater disinfection by combined treatment using multi-soil-layering system and sand filters (MSL–SF): A laboratory pilot study. Ecological Engineering 91 (2016) 294–301, https://doi.org/10.1016/j.ecoleng.2016.02.036.
General comment - edit the list of bibliography according to the guidelines of the journal.
Author Response
Answers to reviewer1 uploaded

Reviewer 2 Report
The work of Roseline et al. is a report of the novel technology multi soil layering (MSL) regarding a first report of the application of the technology on cyanotoxins.
The title is relevant and concise.
The abstract is appealing and fully satisfies the brief reflection of the work that was done.
The introduction is well written, well cited and states the novelty of MSL, and in the last paragraph, the purpose of it in the situation described. Indeed, I could not find such an application nor am I aware of it, therefore the novelty stated is true from my point of view.
The article is well written and well structured, however, minor adjustments are required as far as I can say.
Line 46: please detail the statement regarding the environmental effects and food-chain implications.
Line 83 and 86: a few details about M. aeruginosa and Nostoc would be appreciated.
Line 98-100: the mechanism of phagocytosis / degradation of cyanobacteria should be stated, I would like the authors to feel free to provide as many details as possible regarding this.
Line 104: the reduction of cyanoacteria via the low cost and ecological friendly mechanism of coagulation-flocculation should be detailed
Materials and methods
I appreciate the fact the hour was noted as water O2 saturation varies during the day.
Line 174: please add the HPLC chromatogram as well as the LC-MS one.
Beside that, everything has scientific soundness and looks well done. Congratulations!
Author Response
Answers to reviewer 2 uploaded
